# Impact of the 4 Helix Model on the Sustainability of Tourism Social Entrepreneurships in Jalisco and Nayarit, Mexico

**Rodrigo Espinoza-Sánchez, Carlos Salvador Peña-Casillas * and José Luis Cornejo-Ortega**

Centro Universitario de la Costa, Universidad de Guadalajara, Puerto Vallarta 44100, Jalisco, Mexico;
rodrigoe@cuc.udg.mx (R.E.-S.); jose.cornejo@cuc.udg.mx (J.L.C.-O.)
* Correspondence: carlos.pcasillas@academicos.udg.mx

**Abstract:** Given the uncertain outlook caused by COVID-19, it is important to carry out a review of the conditions in which the collective enterprises are influenced by the four helix model, specifically those dedicated to the sector most affected by the pandemic, tourism, for which raises the question: What have been the results of the four helix model in the social tourism entrepreneurships (STE) of Jalisco and Nayarit? In addition to: the participation of the actors of the four helix model has contributed to face the repercussions of COVID-19? The objective is to identify stakeholder input from the core elements of the four helix model and sustainability to the STEs during COVID-19. The methodology used was qualitative and involved the comparison of information from 12 key stakeholders from the government, social, academic and private sectors through Atlas.ti-8. Some results indicate that from the perception of the participants interviewed, the COVID-19 crisis has promoted innovation, support, and incentives among the four helixes, in which the STEs have benefited. As conclusions, the four helix model is functional to face the adversities of COVID-19 as long as there is planning within the entrepreneurships and the link with said model helix participants.

**Keywords:** social tourism entrepreneurships; four helix model; COVID-19

## 1. Introduction

Entrepreneurship is one of the activities most promoted by governments and universities internationally, given its universal application to the various scientific and economic branches of knowledge that exist, providing more and better opportunities to those who decide to practice it.

Humanity is currently going through one of the most disruptive situations from a social and environmental point of view since 2008, since the COVID-19 pandemic is an event that has changed every aspect of everyday life, and this is forcing the modification of practices and standards, as well as the interaction of organizations with their consumers, affecting the way products and services are acquired, and how supply chains are provided [1].

These changes do not only affect society, since their repercussions are manifested in various actors or institutions that participate in the development processes of the territories, such as companies, governments and universities, where entrepreneurs participate significantly, since the crisis leaves new opportunities for entrepreneurship to become involved in a sustainable way in terms of resources and waste [2] in meeting these new emerging needs and in the post-COVID-19 recovery processes, which involve some of the benefits already found in entrepreneurial initiatives [3] such as the promotion of innovation, growth and economic well-being [1,4–7].

Tourism has been analyzed as an economic activity that energizes all sectors, since this activity is supported by the movement of people, capital and merchandise throughout the planet, and in this specific case, in the rural sector, by taking advantage of the potential of the natural and cultural resources located in the territory in question, and which, based on their attractiveness, give rise to enterprises [8].

Even though the ventures have opportunities to offer solutions to the emerging needs mentioned above, it must be considered that they have certain disadvantages and risks inherent to their nature in comparison with organizations with a longer track record, due to the fact that they do not have an established business model, have low levels of legitimacy and depend on the cooperation of outsiders [9]. On the other hand, the entrepreneurships have some advantages in periods of crisis, considering their small organizational structure, they tend to be more flexible to opportunities or threats in their environment, which also affects the proximity of decision makers to their customers and other stakeholders that can provide them with valuable market information, useful to overcome crises [1,10–13].

According to Espinoza, Marquez and Chávez [14], tourism is a complex phenomenon that requires for its analysis of various disciplines that allow the approach in an appropriate, thorough and precise manner, which demands care from the moment of theorizing to its application to reality, therefore, the companies or community enterprises dedicated to this activity are so vast that they do not require approaches, but deepening in the investigation of their operating conditions in order to understand them, as well as to generate certain criteria that not only guide the inquiry, but also delimit it to generate valuable knowledge.

Therefore, the development of tourism in rural areas is conceived as a strategy to promote the rooting of the local community in its territory, and as a consequence, strengthen the conservation of its heritage, since the vulnerability of these rural communities in Mexico is due to the synergies brought about by this globalized economy, which generally leads to imbalances and asymmetries among the most unprotected sectors, such as the rural sector through the STEs [15].

For the present research, it is necessary to understand STEs from their bases, firstly, social entrepreneurship, which is understood as individuals or organizations in search of opportunities to improve social conditions through the application of practical, innovative, and sustainable methods [16]. Subsequently, the STEs are identified as a form of collective business organization, for the use of resources located within the territory, whether ejido or communal, and where the administration and operation of these business initiatives strengthen and empower the tenant, responding to the poverty of the agricultural and livestock sector [17], which refers to people in rural areas who engage in tourism and thus complement their traditional activities and satisfy existing needs. Another precise aspect to point out is the meaning of ejido, a concept that refers to a figure of distribution and ownership of communal land for the exclusive use in Mexico, where the community must be consulted for the management of these lands.

Such STEs share essential characteristics with communitarian tourism enterprises [18], which are characterized by emphasizing the achievement of common purposes in communities with a shared history and cultural identity; however, STEs are born from the actions undertaken by the basic core of a collective whose objectives go beyond history and culture, combining these elements with the social and solidarity economy that allows the development of recreational activities in a rural space to increase and develop the social capital of entrepreneurs, and thus provide them with greater confidence so that they can generate local development [14]. Some examples of in-depth research to understand the reality in which STEs live and to generate a real impact on them are carried out by the academic sector with governmental resources [19–26], which manifest a process of human capital formation in the practical field, integrating the resources and capacities of different actors or development helixes in a given region, as in this case are the municipalities of Puerto Vallarta in the state of Jalisco and Bahía de Banderas in the state of Nayarit, Mexico.

The above situation highlights tourism as an activity that can give way to social entrepreneurship, but it should not be lost sight of the fact that other actors participate in this process, whose influence can be explained by the model of the four helixes and sustainability, so the question arises: what are the contributions of governments, businesses, universities and society from the approach of the model of the four helixes and sustainability to the STE in Puerto Vallarta and Bahía de Banderas during COVID-19? To answer this question, a review of these models at a theoretical level, the compilation

of information on the diverse elements that integrate both approaches to representatives of each sector are presented, as well as the discussion of the information, followed by conclusions, implications and future lines of research.

## 2. Theoretical Background

### 2.1. Quadruple-Helix Model

In the environment or field of action of the ventures there are some stakeholders that have a strong influence on the results they obtain, which is explained in a basic way in the triple helix model proposed by Etzkowitz and Leydesdorff [27], in which companies, universities and governments are considered as each helix that develops internally, with the capacity to exchange knowledge, products and services [28]. Sánchez [29] indicates that, in addition to knowledge transfer, the propellers of the model are linked through alliances, support and investment, as well as taxes and incentives, which are some of the core elements of this study, as shown in Figure 1.

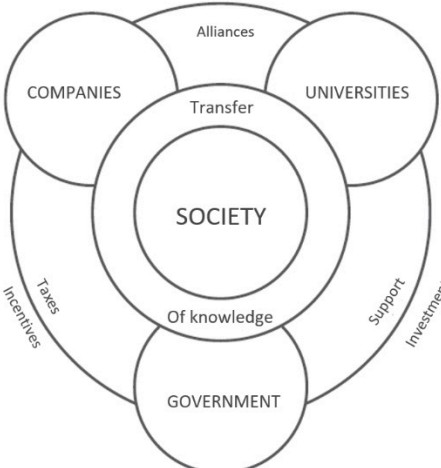

**Figure 1.** The triple helix model. Source: [29].

This triple helix model, driven by the need for innovation, is associated with academic entrepreneurship, since it defines a new dimension of entrepreneurship for universities, in which these and other knowledge-producing organizations play a decisive role [30,31].

The triple helix theory emphasizes the role played by universities, as it considers a proactive participation in open networks and in the generation of innovation that integrates government and industry, which has changed in contrast to the past in which it was limited only to creating knowledge infrastructure and instruments [27,32].

In order to better study development and innovation, the triple helix model evolves by adding a fourth helix representing society, combining among this group of participants the generation of policies, co-creation of knowledge and value, applicable to all types of economies [33–36] where the core elements of alliances, support and investment, taxes and incentives are maintained, and society moves from a central point to a position at a balanced level with the other helixes. At this point, it is possible to ask, what is the relationship between the four helix model with tourism and the STEs? A particular and novel branch of tourism called permatourism is responsible for analyzing the relationship between the actors of the tourism dynamics, both formal and informal, through the creation and promotion of meaningful and functional relationships between them [13,37], and these relationships can be seen from the perspective of the sustainability and the model of the four helixes through the central elements described above. This research is also related to sustainable tourism, since it deals with an ejido that migrates from natural resource depredation activities to their preservation, which is part of its tourism offer that generates economic benefits for the community [38,39].

### 2.1.1. Companies

In order to identify the role played by companies in the quadruple helix model, it should be noted that they exist in different sizes, and their role with society, government and universities is manifested accordingly. Large companies traditionally rely significantly on internal R&D to create new products and services [40,41], therefore, they must constantly collect information from society to generate business knowledge.

This flow of knowledge in the company includes consumer co-creation, information networking, university research grants, contracting with external R&D service providers, IP licensing and crowdsourcing, all leading to innovation, which refers to knowledge flowing through the company through sales, participation in public standardization, corporate business incubation and entrepreneurship, intellectual property licensing, patent sales, and spin-offs [35], elements in which the other helixes of the model can participate through alliances, cooperation and joint ventures [42].

This shows the dependence of large companies on innovation and knowledge generation, so they seek to facilitate these processes, which is not easy for SMEs that have limited resources that can generate difficulties in exchanging external resources [43]. However, alliances, cooperation and networking are common phenomena in SME's, which facilitate their access to downstream markets in the technological field [44]. In general, whether in large companies or in SME's, the literature agrees that establishing alliances and cooperation with the actors of the propellers helps companies to obtain benefits and develop their substantive functions.

### 2.1.2. Government

Through government involvement, collaboration between companies can result in knowledge, products and economic sustainability, because in the past, government functions were limited to regulation, control and standardization, whereas today these functions are shifting to facilitating collaboration between universities, industry and society [35].

The role that governments assume under a joint system with universities, companies and society is in principle to provide the infrastructure that allows the interaction of the different participants for joint innovation, and as an example of this are the intellectual property rights and transactions to share technology [45]. Second, governments stimulate demand and encourage the creation of new markets by building marketing channels, industrial clusters, incubators and strategic alliances with high-tech companies and emerging industries, as well as influencing knowledge through fiscal policies, science and technology policies and capital markets [35,45].

### 2.1.3. Universities

Some studies [1,36] indicate that collaboration between entrepreneurship and co-working fosters and accelerates innovation, bringing mutual benefits to the participating members, and studying these collaborative relationships with the business environment is essential for educational institutions to organize extracurricular learning activities, valuable for students and teachers. The connection with the needs and expectations of the companies, given that it is the supply connection of human capital developed according to the standards of their labor sector, as well as entrepreneurial education and practice can be incentives in the adoption of cultural changes necessary to create new sustainable enterprises.

The traditional contributions that universities make are those of education and fundamental research, knowledge transfer, research and development, according to what society needs [46]. Typical channels of knowledge transfer include publications, conferences and meetings, research, graduate student co-supervisions, consulting, collaborative research [47], patent development, informal communication [48], staff mobility and training [49].

Other authors mention that universities can participate in the role of intermediaries, to bring knowledge producers and users closer together, which generates relationships of trust and commitment [50,51]. Other exploratory studies indicate that a new role for universities is that of trusted intermediaries or centers of open innovation [47]. In addition, universities

participate in the success of ventures through the transfer of know-how, business education, and especially assistance to entrepreneurs [52].

Universities benefit from the four helix model and benefit other participants when they manage to consolidate entrepreneurial education, which does not only refer to specific contents within the study programs or the way in which the content is offered; this concept implies a broader context in which the development and accumulation of skills for life and the way in which these contribute to the quality of life must be taken into account, generating the human and social capital necessary for the transfer of knowledge to take place, with collaboration between the helixes as a key factor [53], as mentioned above, in the study site there are some documented approaches of the university to entrepreneurship to solve problems and address areas of opportunity, which highlights the linkage work required of this type of institutions today [19–26].

### 2.1.4. Society

One of the ways of looking at the participation of society and consumers in society in innovation is through crowdsourcing [35]. For the public sector, governments can use online platforms to gather citizen input and ideas [54], and this technique can be used by companies to feed the information needs of their projects from networks of people through open calls for proposals [55], contributing to decision making related to product development or consumer outreach activities [56].

This indicates that society as seen from the four helix model can be found in consumers, whose role is no longer limited only to purchasing, but who take part in product development [57,58] by providing the necessary information for its creation and development.

On the other hand, society manifests itself in the quadruple helix model as civil society, which includes the media, users, agencies and culture, all as means to drive the innovation process [59]. This model from a sustainability approach applied in the collaboration between industries, governments, universities and society can be used for the design of strategies to achieve a green economy [60]. In the study region, the possibilities in terms of tourism are diverse, since this well-planned activity benefits society, due to the fact that new and differentiated tourism products are possible, where intelligent tourism can be developed, in such a way as to promote development through collaborative participation for greater equity and sustainability [61].

### 2.2. Aspects of Sustainability

Nowadays, sustainability has become a critical issue due to a worldwide concern for the impact of companies on natural resources, the environment and the society that is affected by these repercussions, where three main axes can be appreciated: economic, environmental and social [62].

From this arises a new entrepreneurial perspective in which sustainability and the four helix model can generate sustainable ventures for the academic sector [63], as each participant has multiple and simultaneous roles. Therefore, the need for innovation that gives rise to the quadruple helix model implies that the public sector, academia, industry and the social sector collaborate to generate structural changes beyond those that can be achieved individually [36].

Both the field of sustainability and its factors and the four helix model are closely related aspects, with a limited number of practical research that manages to combine both models for the study of the conditions in which enterprises operate, which is of utmost importance in a crisis scenario, so that the flexible, dynamic and open collaboration between the helixes in a sustainable way brings new possibilities in terms of social needs, culture creation, political support, green economy, responsibility and technological advancement [35].

For the present research, the five dimensions of sustainability at the operational level proposed by [64] are considered, as follows:

- Social: it aims at improving the quality of life, including participation in decision-making processes, and activities related to the assistance to people.
- Economic: through growth with equity and efficiency, involving activities that generate economic values or monetary repercussions.
- Ecological: in terms of ecosystem conservation and integrity, including activities and efforts to preserve and restore the environment.
- Cultural: out of respect for diversity, as well as the preservation and promotion of traditions, folklore and gastronomy.
- Territorial: the search for spatial balance in development, and actions that have an impact on the use, control and exploitation of space and infrastructure.

As observed above, sustainability applied to the participants of the model of the four helixes allows explaining some of its functions at a theoretical level, which may be different when it is landed in the practical field, affecting the STEs, therefore, again the question that guides the present study is posed as follows: What are the contributions of governments, businesses, universities and society from the approach of the model of the four helixes and sustainability to the STE in Puerto Vallarta and Bahía de Banderas during COVID-19? The context of application of the study and the methodology by which information was collected from representatives of each of the helixes of the quadruple helix model is explained below.

## 3. Materials and Methods

### 3.1. Setting the Context: The Impact of COVID-19 on Tourism in Mexico

As mentioned in previous sections, the crisis generated by COVID-19 is something unprecedented that has generated historical repercussions in the various sectors of the economy and in the way in which people live on a daily basis, and entrepreneurship in the tourism sector was not exempt, but this activity was one of the most vulnerable to this phenomenon, due to its high dependence on the mobility of people.

Figure 2 shows that the level of variation in the arrival of international visitors had not presented a value higher than 8.9% until 2019, however, by 2020 there was a drop due to the COVID-19 crisis and the mobility restrictions implemented by many countries, which resulted in a drop of 47.5% [65], and this impacted all economic sectors, especially tourism, which had to stop travel for leisure and recreation, prioritizing travel for humanitarian causes.

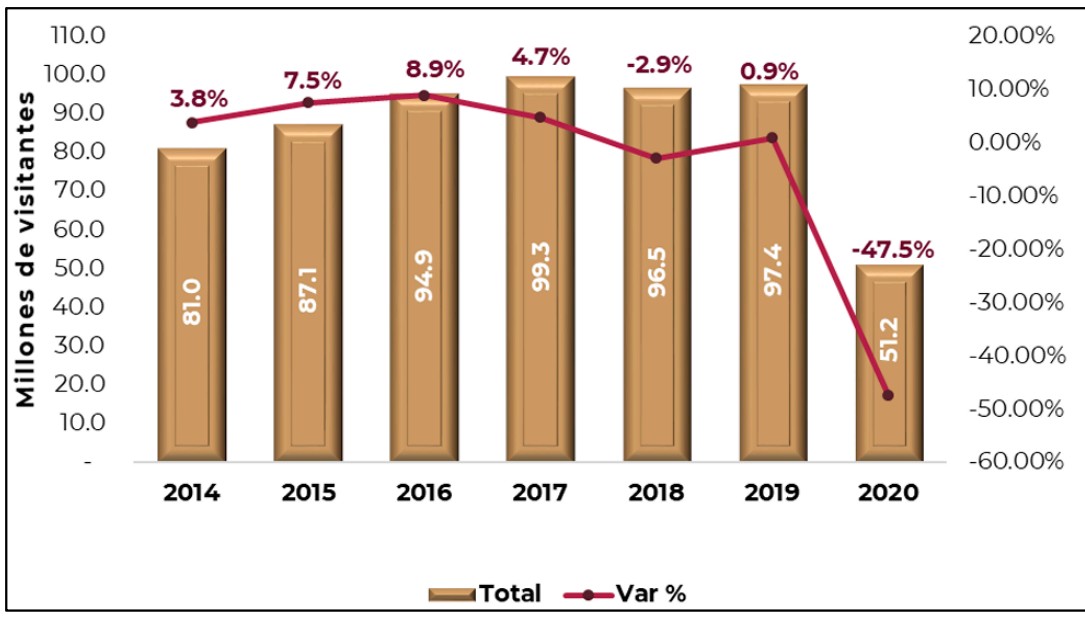

**Figure 2.** Comparison of international visitor arrivals to Mexico 2014–2020. Source: [65].

This decrease affected to all organizations without distinction, imposing important economic pressures on businesses and even on the three levels of government (municipal, state and federal) due to a lower economic flow, for which reason some tax payment deferral alternatives were implemented, as well as programs to protect family savings by giving money to the neediest families.

The governments of Jalisco and Nayarit established some measures applicable to the regions studied in Puerto Vallarta and Bahía de Banderas (Figure 3) to reduce infection rates, such as the prohibition of some recreational activities, as well as the establishment of limits on the hours of operation of most businesses, except those dedicated to the provision of medicine, sanitary supplies or medical care. Another of the most significant measures was the total halt to face-to-face educational activities at all levels, where education was carried out through television programs and virtual modalities in the case of high school and higher education.

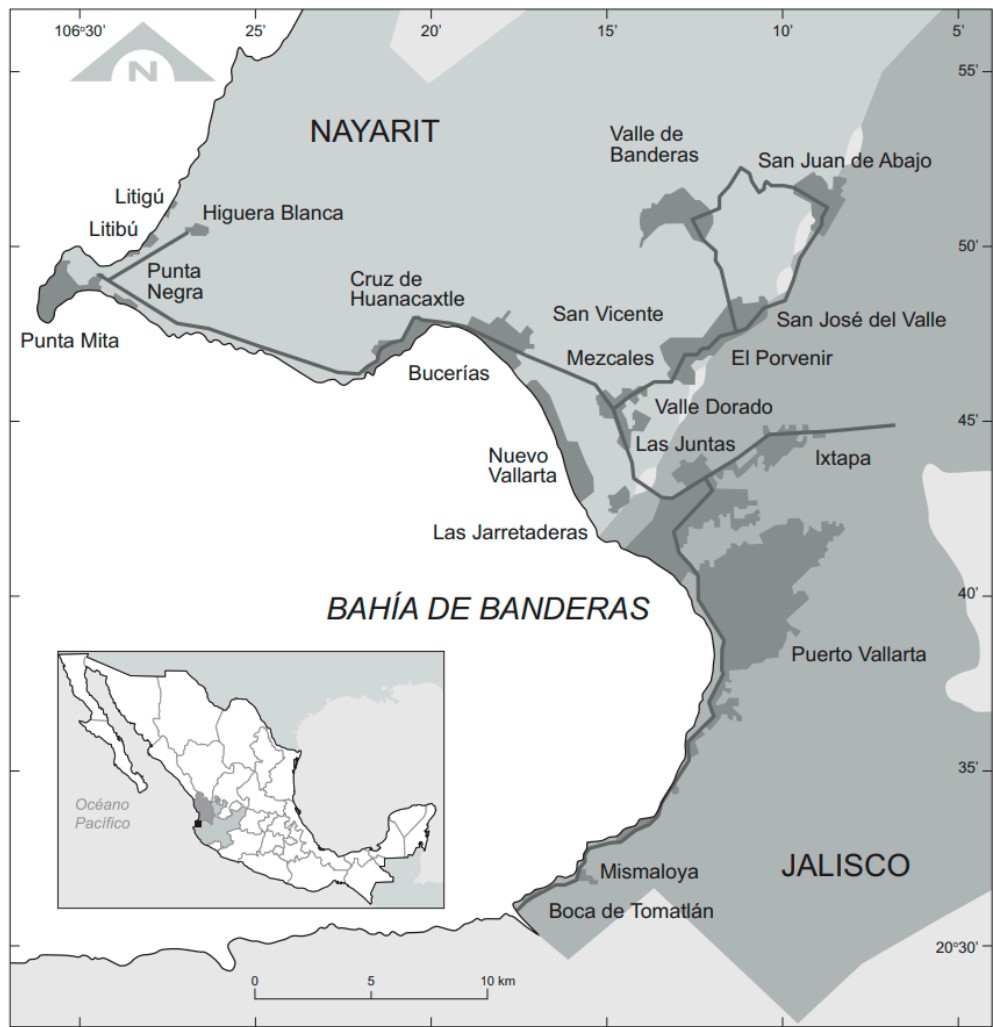

**Figure 3.** Map of Puerto Vallarta, Jalisco and Bahía de Banderas, Nayarit. Source: [66].

One of the most significant cases of STEs in the study region is found in the ejido space called El Jorullo (Figure 4), which is home to three important references in entrepreneurship that are documented in graduate work or theses of graduate students [21–24], who based their development on addressing an area of opportunity or solving a problem present in these ventures, which shows the existing link between the university and these initiatives.

This ejidal space is special because it is home to a group of farmers who became entrepreneurs, motivated by important deficiencies in their territory, which due to physical

reasons did not allow them to dedicate themselves fully to extensive agricultural activities, so that necessity forced them to cut down trees and hunt species to subsist. Faced with this precariousness, a small group of ejidatarios sought alternatives that would allow them to improve their living conditions, however, due to the legal status of their lands, it was not possible for them to apply for financing for their ideas, and it was not until an investor believed in them and provided them with the necessary financing to venture into ecotourism activities in which they could take advantage of the benefits of their territory through the tertiary sector [68]. This, as well as a long "trial and error" trajectory in ecotourism, allowed these farmers to become entrepreneurs through the development of management skills in a project called Canopy River, which, under the demonstration effect, has allowed the emergence of two more ventures in El Jorullo. This new approach allows entrepreneurs to operate within the framework of sustainable tourism, moving from the depredation of natural resources to their conservation, including local cultural resources, generating sustainable tourism [38,39].

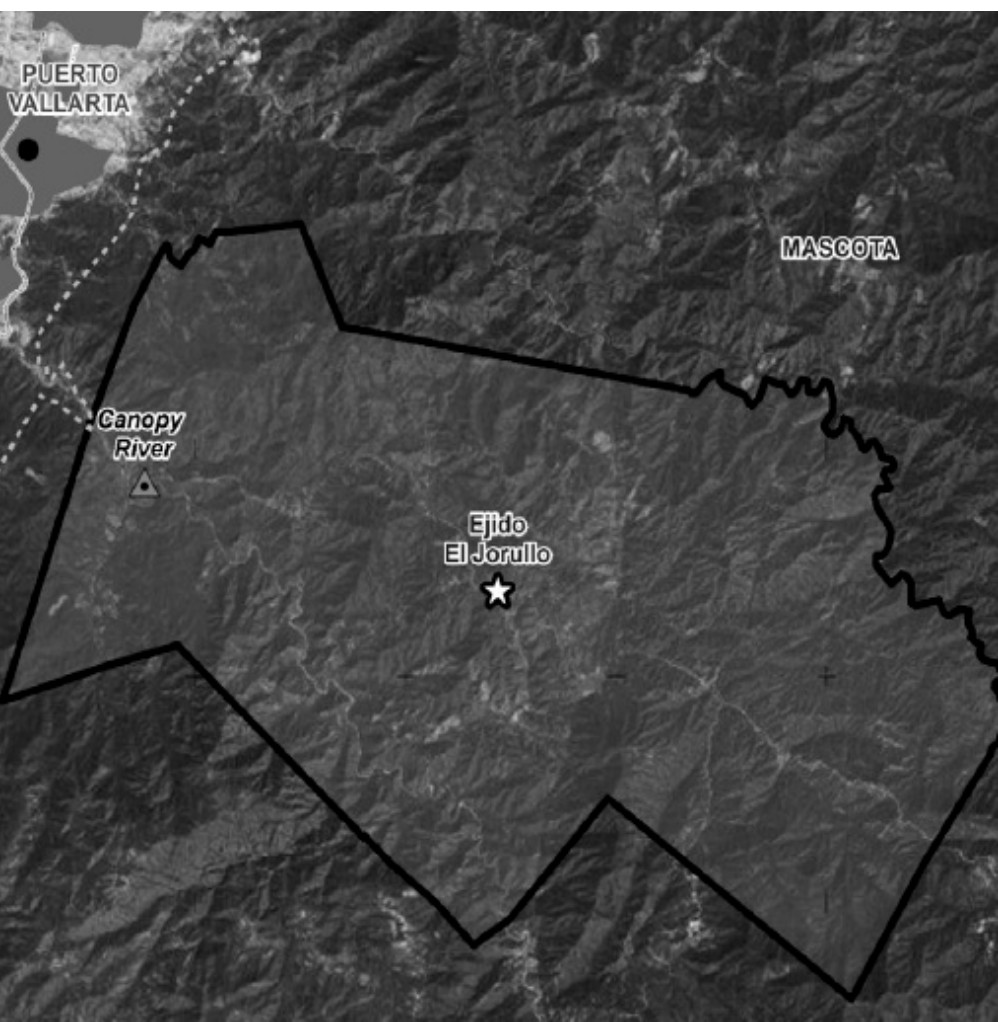

**Figure 4.** Map of the ejido El Jorullo. Source: [67].

*3.2. Research Methodology*

Based on the characteristics of the observed phenomenon, in this case the STEs, and the conditions of their environment, in accordance with the provisions of Espinoza, Chávez and Marquez [14], qualitative research was chosen to allow a deeper understanding of the interaction that takes place in the quadruple helix whose operation is based on human interaction [69].

For this, semi-structured surveys were used, consisting of 10 questions for each of the 12 key actors to answer openly, giving them the opportunity to mention relationships between the topics studied and the other actors or helixes, until data saturation was obtained. The topics addressed were posed with respect to the participation of the sector itself in each during the COVID-19 crisis period, as follows:

1.  A total of five open-ended questions corresponding to the topic of sustainability mentioned by [64], which are the social, economic, ecological, cultural and territorial spheres;
2.  A total of four open-ended questions correspond to helix linkages [29] on innovation, partnerships, taxes and incentives, as well as support and investment;
3.  One open-ended question related to the particular impacts of COVID-19 on the participant's sector.

To guarantee the internal validity of the study, all the information provided by the informants was recorded and transcribed. A content analysis was applied to the data collected [70], and the information provided by representatives of the social sector, government, universities and companies was structured and classified using Atlas.ti-8 software, in order to understand the differences experienced by each of them in the same crisis context and to draw conclusions.

*3.3. Participants*

To carry out the study, a non-probabilistic purposive sampling with key actors related to tourism in the study area was used. For the selection, the type of participant was first considered according to the four helix model, so that each helix of the model was represented (business, society, government and university), and then their type of participation was coded according to the state of incidence, in which there were only two types, Jalisco and Nayarit. In total, in-depth responses were obtained from 12 participants (Table 1).

**Table 1.** Informants of sectors by state.

| Informants | Code |
| :---: | :---: |
| Entrepreneur, 1, social sector | Jal1 |
| Entrepreneur, 2, social sector | Jal2 |
| Entrepreneur, 3, social sector | Jal3 |
| Regional social actor, 4, social sector | Nay1 |
| Representative of the municipal government, 5, government | Nay2 |
| Professor and university researcher, 6, university | Jal4 |
| Professor and university researcher, 7, university | Nay3 |
| Professor and university researcher, 8, university | Jal5 |
| Representative of the municipal government, 9, government | Nay4 |
| Representative of the municipal government, 10, government | Jal6 |
| Representative of the municipal government, 11, government | Jal7 |
| Company representative, 12, companies | Jal8 |

## 4. Results

The results of the information gathering were organized in levels that allow understanding the overview of entrepreneurship in the studied territory, starting from the relationship that each one of the participants or helixes establishes with the tourism social enterprises, and in a second level the way in which these development actors contribute to the connecting aspects of the four helixes model and also on sustainability factors, as well as their repercussions on the COVID-19 theme (Figure 5).

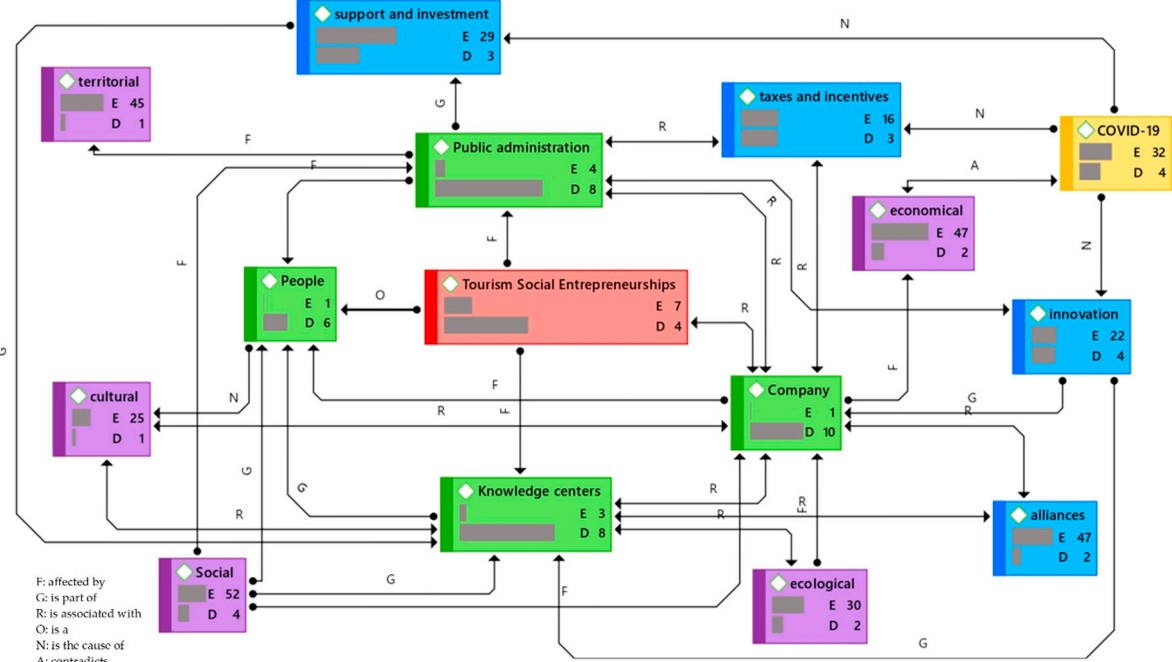

**Figure 5.** Network of codes in Puerto Vallarta and Banderas Bay on sustainability, the four helix model and COVID-19. The central theme of the study appears in red; the participants under the four helix model classification in green; the helix interaction factors in blue; the dimensions of sustainability in purple; the COVID-19 factor in yellow. The Letters E and D within the codes indicate the following: rooting (E) or the number of citations connected to that code, which refers to saturation as part of the content analysis; and density (D) indicating the number of connections to other citations. Source: authors. Network created with Atlas.ti-8 data analysis software. Source: Authors.

Fifteen codes were identified according to the information collected in the content analysis of the informants, which allowed the generation of codes and the establishment of the type of nodes between each element in relation to the actors of the four helix model, as well as their connection with COVID-19. In relation to the context of entrepreneurship in the region studied, the presence in the discourses of information with a high degree of rooting was manifested [71] oriented mainly to the territorial, economic, social and alliances aspects. The main contributions and repercussions for each actor of the quadruple helix model are presented in relation to the categories analyzed.

*4.1. People*

Entrepreneurship is an activity designed to respond to the needs of its members, and in this case, there are efforts on the part of the entrepreneurs for development, in this case oriented to the formation of human capital in terms of culture, language skills, as well as the promotion of infrastructure for the promotion of health:

> 1:24. We also participate in trainings, in different trainings in the community that we have supported within the community together with the ejido, for example we have put together a little music group, we have contributed in folkloric ballet, in other types of English trainings for the youth of the region, just to mention a few, also in the health house we have contributed in the health house and in the multipurpose courts (Jal1).

> On the other hand, the social entrepreneurship sector has worked in alliances, but they still consider them insufficient, since "So far, the alliances we have had so far have been fine. Maybe we still need to knock on more doors to have more" (Jal3), and even other participants in the social sector find this factor weak, "Each sector

or each productive agent, all operate on their own. Some are establishments, and there is no integration" (Nay1).

The cultural aspect is manifested as part of people's daily lives; it is an issue that the participating entrepreneurs perceive from different and varied points of view, where there is an interest in preservation:

> 1:28. To have our cultures, our traditions, we are like country people, Mexican people, but we like to live, to live the traditions, the cultures, and the folklore, and we even handle it here in some events that we have. This is the folklore of Mexico, not to lose it, and also the gastronomy (Jal1).

The ecological part for the entrepreneurs is a change in the traditional ways of life and production, because they were able to find a new way of obtaining benefits without excessive exploitation schemes, and they have even conducted restoration work, visualizing the preservation of the environment as a means of subsistence:

> 1:27. Today we are looking for friendly projects, before you can see that we were dedicated to the countryside, that is to say, to destroy today to take care, to preserve what we have, what we sell are landscapes. So the better the conditions of the landscapes, the more visitors we can have (Jal1).

On the other hand, the COVID-19 situation has had a significant impact on the economic environment, where activity levels have dropped considerably: The economic environment is critical right now, it is low right now. I believe we are at 10% (Jal3).

> Faced with these negative economic impacts, entrepreneurs have been supported by the government in terms of taxes: The Government has made it easier for us to pay by extending payment terms, up to one year without paying interest, for example, there has been more flexibility (Jal1); But last year we did receive a little support (Jal3).

Support for entrepreneurs, especially in times of crisis, is of vital importance because they themselves imply benefits even of a territorial nature, through some additional actions to improve their living conditions:

> 1:29. We have even done works for the benefit of the territory, for the benefit of the environment, soil retention works, we have soil retention works at ground level, we have not done roadways. Works will work to take care of. We have done reforestation.

### 4.2. Public Administration

The governmental entities addressed recognize the importance of partnerships, stating that "with the global economic issue we consider that we need to work as a team, those who work alone will practically disappear" (Nay4). Knowledge in innovation processes, training and governance is also key, as they have established alliances with knowledge centers such as universities:

> 5:13. Well, we have alliances, recently with the ITAM, with the University of Guadalajara, with the business sector, to mention a few others regarding training, management, there are countless alliances that have been made throughout this administration (Nay2).

Regarding the COVID-19 situation, the government sector implemented some joint support programs with businesses for individuals and families, "the government gave a food voucher of 400 pesos, three times during the pandemic in 2020 that are 400 pesos, reactivating the economy to the corner stores, they registered" (Nay2).

Another of the municipal government's infection prevention activities is based on innovation through technology, by transmitting prevention protocols and information from higher levels of government, as well as control functions:

> 10:4 We then followed those indications and applied them, and on the subject of protocols, we were making small businessmen aware of them, we even shared information with Riviera Nayarit for that, the same on the subject of paperwork, we enabled some things on our web page, so that people did not have to bring a document per se, but that it was digital (Nay4).

> 13:4 As a municipal government within our functions we are not the rectors or authority in the health issue, but we have to guarantee that the restrictions established by the state government are complied with, to guarantee health, to cut the chain of contagion (Jal7).

In the ecological sense, the municipal representatives emphasize the importance of the environment, "today the government's decisions must be focused on policies that have to do with the care of the environment" (Jal7); and this is identified in a new vision of tourism called alternative tourism, which combines economic aspects with the current needs of environmental care:

> 12:5 The new modality of tourism is one of respect and care for the environment, for which we have the enormous responsibility to take care of our mountains, seas, rivers, flora and fauna, to integrate them to it in a respectful way, although the new tourist practices in the so-called alternative tourism (Jal6).

On the economic front, the municipal government highlights tourism promotion activities through social networks and official websites, thus promoting greater economic spillover and opportunities for business and entrepreneurship:

> 10:6. And in the economic sense, in the sense of promotion, we try in our official pages or official social networks to promote everything, from tourist developments, tourist attractions and all the tourism enterprises, everything we have as a tourist vocation, but encompassed in a whole, in a municipality (Nay4).

> Government innovation has focused on revenue collection and support by designing platforms that allow virtual interaction, "the online payment of the property tax and certain social support issues that the system also has so that you can register to receive some type of support (Nay2); or the use of other tools, such as the codes QR, "we started to use QR codes, and those QR codes guide us directly to what we want" (Nay4).

Some tax supports were implemented during the crisis season, "in taxes for the first three months there is a discount to help the economy of the population" (Nay2), "The reduction or reduction in the payment of their operating licenses, in their construction licenses, in everything they have to pay in terms of municipal taxes" (Nay4), and this also implies that mechanisms must be generated to improve collection efficiency, which will result in better infrastructure conditions and public services offered to people, enterprises and businesses:

> 13:18. There has been an increase in the collection of property tax, and this in the end is to the benefit of the citizens because there are greater resources, greater capacity to invest and improve the infrastructure when the resources are exercised in a transparent and responsible manner (Jal7).

*4.3. Knowledge Centers*

For recognized knowledge centers in universities, alliances are an important factor because "that it is the way in which the university makes its impact, if it does not do it through alliances, it does not ensure the current validity of knowledge (Jal4). The resources available to universities can foster partnerships:

> 9:6. There must be a greater participation with the university, taking advantage of its laboratories, taking advantage of its researchers, taking advantage of its knowledge, for the development of more contributions (Jal5).

These resources, applied to knowledge, allow the university figure to gain a liaison role between government and society, since "then the university with the research, the generation of new knowledge is bringing the government closer, is bringing society closer to know more about what is needed . . . " (Jal5).

Another role played by the university is related to COVID-19, "support with information systems, for the application of diagnostic tests, in the development of research protocols" (Jal4), "inviting the students to assume with responsibility and commitment that the situation we are going through" (Nay3).

In the area of culture, the ways in which this manifests itself in the university are as follows " . . . through all artistic expressions, the promotion of interaction with other cultures and cities of the university, then we do it through art, we do it through festivals" (Jal4), and in the case of the university community, the preservation of this element of sustainability is sought:

> 8:9 What we as an institution have to do is to seek the means and, above all, the arguments and attributes that can provide students with greater tools for the conservation and management of heritage in all aspects, be it natural, cultural, mixed, intangible, oral or documented heritage, all of which contribute to the safeguarding of these resources, above all training (Nay3).

On the other hand, the contribution of the knowledge centers addressed is oriented to the development of people, in this case students, in order to " . . . to have a positive impact on the economic, on a more balanced development of all sectors would depend on the human capital itself, because human capital is the resource that combined with the other economic resources produces the results" (Jal4). For universities, the theme of innovation turns out to be one of the strongest ones, present in "knowledge, skills, abilities, skills, and I also believe that the involvement of professional activities in young people, managing them in the industrial field" (Jal5). Innovation also allows the university to link to entrepreneurship, because:

> 6:15 We can talk about innovation and entrepreneurship centers that exist around the entire university network, which are precisely the way, they should be channelers of support, of investment promotion, maybe there is a little more action needed to promote or facilitate investment (Jal4).

### 4.4. Company

The business sector was significantly affected by the repercussions of COVID-19, mainly for revenues in contrast to a 1-year pre-crisis scenario, taking into account the following:

> 14:18. The crisis that we are experiencing right now, where everyone is talking about the fact that the year 2020 compared to 2019 in general is representing a 50%, 70% drop in income, as an average, because there are companies that reach up to 80% (Jal8).

These repercussions generated a response, in the first place, of a technological nature, as companies were able to innovate in this sense to promote contents for the prevention of contagion: "take advantage of technology through awareness and educational videos, which we ask the employees to transmit them in their homes, with their partners, with their children, and also with all the people who have influence" (Jal8).

Other important actions of the companies in the cultural element within sustainability are some programs for the preservation and promotion of culture among the employees themselves, which consist of:

> 14:10. We have a program where periodically we make a state of the republic where the collaborators who are from that state of the republic, set up the presentation and talk to the other collaborators about their state but they also prepare the typical food of that region, and they bring from their homes all those cultural elements that are important for them (Jal8).

For the ecological issue, even though there are some activities to favor the preservation and restoration of the environment, the business sector recognizes that there are still areas of opportunity in this regard, due to the fact that:

> 14:14. . . . the campaigns of care and cleaning of beaches and rivers, I believe that here there is a great advance but it is not the only thing, then I believe that here we have to work more on the part of the businessmen (Jal8).

## 5. Discussion

Both in the area of sustainability from the factors proposed by [64] as the four helix model [27,28,34–36,63] with their respective participants and elements are theoretical aspects of great value and complexity that allow us to better study the interrelation of diverse tourism related participants in a dynamic innovation and development system, in which all the actors respond and contribute in different ways to the pressure of the crisis caused by COVID-19. In the tourism regions of Puerto Vallarta and Bahía de Banderas, corresponding to the states of Jalisco and Nayarit respectively, it could be seen that even with participants of the same nature, the approach towards sustainability, innovation and actions in the face of COVID-19 were dissimilar.

According to what has been observed, the ventures have areas of opportunity for the establishment of alliances, cooperation and joint ventures that allow them to exchange resources [42,43] because the informants believe that integration between productive sectors needs to be strengthened.

On the part of the government, it was observed that in the face of the crisis, its efforts are not focused on the search for new markets [35,45], but on the implementation of various strategies, tax incentives, and support activities through promotion to maintain existing markets in the face of the economic shocks that the current crisis is inflicting on the social and business sector, as well as some control activities on the issue of the spread of contagions.

Universities in this region have been able to adapt and channel their resources to meet the needs of society [46], although COVID-19's situation has diminished its personnel mobility and training capabilities [49] by forcing a transition from a face-to-face to a virtual education system and affecting its typical knowledge transfer channels [47], but even so, education has been maintained, promoting aspects of ecological care and the establishment of alliances.

In the corporate sector of tourism, the traditional functions based on knowledge flows [35] have had to be modified, emphasizing in this new panorama of crisis towards social responsibility, where the human element is manifested as one of the main and most vulnerable resources, and the subject of innovation has been focused on its care, so that technologies act as factors for the prevention of COVID-19 contagion, and some strategies for the preservation of the cultural environment of its workers were also evidenced.

## 6. Conclusions, Implications and Future Lines of Research

### 6.1. Conclusions

First of all, the STEs studied have been generated due to the onslaught of the prevailing global economic model that has led to increased poverty in rural areas, and this has led to tourism becoming a palliative to try to improve the living conditions of the local communities in which these enterprises are located.

These STEs have also managed to preserve their cultural identity and recover the flora and fauna in their territory, an aspect that implies many benefits for the quality of life of the communities and, at the same time, is aligned with the objectives of sustainable

development thus contributing to the practice of this four helix model, which is implicit in the sustainability.

It is very important that theory be applied in the practical field to identify, explain and act in the face of adverse phenomena in different realities and contexts, as in this case, the actors of the quadruple helix model and the various exchanges they carry out in terms of innovation, development and sustainability, where several of these elements are linked in the various actions carried out, especially in a crisis scenario such as COVID-19, which has even worked to reduce the predominant interest in the economic aspect and promote other elements of development and sustainability such as innovation in the case of the use of technologies, taxes at the municipal level for those businesses and population affected by the pandemic and even support in the maturity of credits.

These actions implemented by the actors studied show an important degree of empathy and responsibility in difficult times, so it was appreciated that the networks between the nodes of the network were strengthened, and each effort or action taken to improve in relation to any of the factors studied has repercussions on the central theme of the study, which in this case is entrepreneurship.

It should be noted that the research activities carried out by the knowledge centers in the practical field of human capital formation is of vital importance, since it generates a beneficial relationship for the students in training, who generate experience for themselves and for the universities with respect to research and improvement proposals for the generation of more value in the activities of the enterprises. This, together with the technical, intellectual and infrastructural resources that higher education centers make available to the other development propellers, become spaces that encourage collaboration and integration between sectors that the social sector considers necessary.

Therefore, understanding the complexity of the existing relationships between the four helixes studied here and the interweaving they perform in terms of innovation and sustainability factors is relevant in terms of knowledge generation, since it raises the scenario in which the ventures must participate and develop, which is not perfect, and has areas for improvement described above that generate better opportunities for success for the entrepreneurial community.

Finally, tourism is seen as a way in which the entrepreneurs have managed to connect, with the objective of improving their quality of life and those of their families, and this has led them to move in an organized way to a sector that they did not know as farmers and that they now consider of great value for their future generations, and in the end, these actions that they have performed are conceived as innovation through tourism, all this possible thanks to the other helixes of the model.

### 6.2. Implications

This study contributes by observing two perspectives of the models widely discussed and applied individually, such as the model of the four helixes and the dimensions of sustainability, and applying them together in the practical field to identify the conditions in which the STEs in the study area are found during the COVID-19 crisis before the participation of universities, government and private companies.

This approach to the application of the theoretical models described above made it possible to identify the existing links between the participants related to the tourism dynamics studied during the pandemic, and to understand this is necessary in the training of university students who will be part of society, businesses, and governments. It is also recognized that this type of research can be the preamble for the design of improvement proposals and policies for each sector, as well as subsequent research that allows observing the evolution of these relationships in a post-COVID-19 scenario.

The limits of the research are found in the restricted availability and number of the respondents due to COVID-19, as well as the method used, taking into account that it was qualitative research, the results were focused on the production of knowledge, not

on its validation, so they cannot be generalized to other realities different from that of the present study.

*6.3. Future Lines of Research*

Future lines of research include the application of this contrast in other locations and with larger participants, as well as a quantitative study that would allow the identification of correlations between the factors discussed here. An additional study could be carried out by contrasting the conditions of the ventures between outgoing and incoming governments.

**Author Contributions:** All authors had equal and significant contributions to this work. Each author contributed to conceptualization, methodology, software, validation, formal analysis, investigation, resources, data curation, writing, review and editing. All authors have read and agreed to the published version of the manuscript.

**Funding:** The APC was funded by Universidad de Guadalajara. Funding number 261595.

**Institutional Review Board Statement:** The study was conducted in accordance with the Declaration of Helsinki, and approved by Ethics Committee of CENTRO UNIVERSITARIO DE LA COSTA of UNIVERSIDAD DE GUADALAJARA (Protocol code CUCPV/SA/CI/06/2021, 19 October 2021).

**Informed Consent Statement:** Informed consent was obtained from all subjects involved in the study.

**Acknowledgments:** All the authors wish to thank the participants for their answers and support in achieving this research.

**Conflicts of Interest:** The authors declare no conflict of interest.

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
