# Peer review of "Impact of the 4 Helix Model on the Sustainability of Tourism Social Entrepreneurships in Jalisco and Nayarit, Mexico"

_sustainability, doi:10.3390/su14020636_

Round 1

Reviewer 1 Report

The aim of the paper is not clear. The title suggested the focus on Tourism Social Entrepreneurship (STE), but this concept can be found only two times (one mention in the abstract and another mention at row 221). Not even social entrepreneurship concept is developed somehow in the Introduction and Theoretical Background sections. Moreover, there no any mention about tourism in the Introduction section. The unclear purpose of the research can be discovered even from the abstract where is stated that “Some results indicate that in the entrepreneurships the participation of the official sector and the university has been different, as well as the benefits have been differentiated to all participants”.

Some references about tourism (but hard to be framed into a discussion of tourism social entrepreneurship) can be found in the section between lines 346-361. As the result, all the work of interviewing 12 key stakeholders does not have any reason. This is the main limitation of the study. The number of respondents is a limitation that authors have acknowledged.

Author Response

  1. The aim of the paper is not clear. The title suggested the focus on Tourism Social Entrepreneurship (STE), but this concept can be found only two times (one mention in the abstract and another mention at row 221). Not even social entrepreneurship concept is developed somehow in the Introduction and Theoretical Background sections. Response: A conceptualization of social enterprises and STE and their elements was added, and these concepts were discussed in contrast to that of communitarian tourism enterprises.
  2. The unclear purpose of the research can be discovered even from the abstract where is stated that “Some results indicate that in the entrepreneurships the participation of the official sector and the university has been different, as well as the benefits have been differentiated to all participants”. Response: The quoted text was replaced by a clearer one: "Some results indicate that from the perception of the participants interviewed, the COVID-19 crisis has promoted innovation, support, and incentives among the 4 helixes, in which the STEs have benefited".
  3. Moreover, there no any mention about tourism in the Introduction section. Some references about tourism (but hard to be framed into a discussion of tourism social entrepreneurship) can be found in the section between lines 346-361. As the result, all the work of interviewing 12 key stakeholders does not have any reason. Response: A paragraph was added in the introduction where tourism is related to the study and the research question. Some types of tourism related to the research are referred to in lines 119-130. 
  4. This is the main limitation of the study. The number of respondents is a limitation that authors have acknowledged. Response: The number of respondents was recognized as a limitation of the study.

Reviewer 2 Report

The content of the article is incompatible with its title. There is practically no thread in the text (apart from the mention of the promotion of tourism and the protection of cultural and natural landscapes) about sustainable tourism! I have the impression that (with reference to the presented results), it is not possible to improve the article without focusing research on sustainable tourism.

Author Response

  1. The content of the article is incompatible with its title. There is practically no thread in the text (apart from the mention of the promotion of tourism and the protection of cultural and natural landscapes) about sustainable tourism!. Response: The concept of sustainable tourism was added in the Theoretical Background and related to STEs in the region in lines 302-308, explaining how one community of ventures improved through this type of tourism.
  2. Some comments in the PDF were clarified, as well as the authors were added within the citations that required it. Added content on social tourism entrepreneurships in the theoretical background. The participants information was expanded. The results have few direct references to tourism because the study categories analyzed are closely related to it, as explained in the theoretical background.
  3. The records asked in line 279 are specific to Atlas ti, used in other investigations of this type.
  4. Greater emphasis is placed on tourism in the discussion and conclusion sections.

Reviewer 3 Report

The article "Impact of the 4-helix model on the sustainability of Tourism 2 Social Entrepreneurs in Jalisco and Nayarit, Mexico" deserved the reviewer's best appreciation and attention.

Congratulations on the article! The topic is very relevant, interesting and addresses the purpose.

There should be more emphasis:

What does the study do?

What did the study bring us again?

What contribution(s)?

Consider reviewing the objective of the investigation making it clearer

Perhaps a table with the methodological steps for better understanding.

Author Response

The first questions in your comments are resolved with a small modification to the implications part. 

"Perhaps a table with the methodological steps for better understanding." Response: At the end of introduction was added a paragraph of the research question and the way to resolve it in a clearer way. A table was not used for space reasons.

Thank you very much for your comments.

Round 2

Reviewer 1 Report

The new version was significantly improved.

I suggest authors to explain the meaning of the word “ejido” as it is quite frequently used. The 6.2 Implications section of the article is still treated insufficiently.

Author Response

The meaning of ejido was added. Improved the implications section.

Reviewer 2 Report

Dear authors, the fact that you use the word "tourism" three times in "Discussion" does not make it an article on tourism. In the summary, the word tourism does not appear even once (still - I wrote about it earlier). In my opinion, this article should not be published in its current form, because it is inconsistent with its topic, and the research, and especially the conclusions, do not have much to do with tourism.

Author Response

Greater emphasis was placed on tourism in the document. The paper deals with tourism social entrepreneurships.